# Functional asymmetry and electron flow in the bovine respirasome

Joana S Sousa, Deryck J Mills, Janet Vonck, Werner Kühlbrandt*

Department of Structural Biology, Max Planck Institute of Biophysics, Frankfurt, Germany

**Abstract** Respirasomes are macromolecular assemblies of the respiratory chain complexes I, III and IV in the inner mitochondrial membrane. We determined the structure of supercomplex $I_1III_2IV_1$ from bovine heart mitochondria by cryo-EM at 9 Å resolution. Most protein-protein contacts between complex I, III and IV in the membrane are mediated by supernumerary subunits. Of the two Rieske iron-sulfur cluster domains in the complex III dimer, one is resolved, indicating that this domain is immobile and unable to transfer electrons. The central position of the active complex III monomer between complex I and IV in the respirasome is optimal for accepting reduced quinone from complex I over a short diffusion distance of 11 nm, and delivering reduced cytochrome $c$ to complex IV. The functional asymmetry of complex III provides strong evidence for directed electron flow from complex I to complex IV through the active complex III monomer in the mammalian supercomplex.

*For correspondence: werner. kuehlbrandt@biophys.mpg.de

## Introduction

Mitochondria are intricate membrane organelles found in virtually all eukaryotic cells, where they serve a number of essential physiological functions. Their central role is to provide energy to the cell in the form of ATP by oxidative phosphorylation. The mitochondrial respiratory chain consists of four complexes (I–IV), which transfer electrons from NADH and succinate to molecular oxygen. A part of the energy gained in electron transfer is used to pump protons across the inner mitochondrial membrane. The resulting proton gradient is utilized by the mitochondrial ATP synthase to generate ATP. The respiratory chain complexes reside mostly, if not entirely, in the mitochondrial cristae (*Kühlbrandt, 2015*), which are deep invaginations of the inner membrane into the matrix.

NADH:ubiquinone oxidoreductase, also known as complex I, is the largest assembly in the electron transfer chain. Mammalian complex I comprises 44 different subunits, including two copies of subunit SDAP, and therefore consists of a total of 45 subunits (*Vinothkumar et al., 2014*). The 14 core subunits are conserved from prokaryotes to mammals (*Walker, 1992*). The characteristic L-shape of complex I arises from the association of three different units. The dehydrogenase and hydrogenase-like units constitute the matrix arm and are responsible for the transfer of electrons from NADH to ubiquinone (*Sazanov and Hinchliffe, 2006*). As a third unit, the membrane-embedded transporter assembly pumps four protons from the matrix to the cristae lumen per catalytic cycle (*Galkin and Terenetskaya, 1999*; *Leif et al., 1995*). Succinate:ubiquinone oxidoreductase, or complex II, is the only complex in the electron transfer chain that does not translocate protons, but merely feeds electrons into the process. The cytochrome $bc_1$ complex (complex III) is a symmetrical dimer, with cytochrome $b$, the Rieske iron-sulfur protein and cytochrome $c_1$ as core subunits (*Yang and Trumpower, 1986*). The Rieske subunit extends across both monomers, stabilizing the dimer that is essential for function. In mammals, complex III contains a total of 11 subunits per monomer, of which eight are supernumerary (*Schägger et al., 1986*). Complex III transfers electrons from ubiquinone to cytochrome $c$, a small soluble electron carrier protein in the cristae lumen. Finally,

cytochrome *c* oxidase, also known as complex IV, transfers electrons from cytochrome *c* and catalyzes the reduction of molecular oxygen to water. Mammalian complex IV has three core subunits (COX1, COX2 and COX3) and 14 subunits in total (*Balsa et al., 2012*; *Kadenbach et al., 1983*).

The structural organization of the complexes that carry out oxidative phosphorylation in the inner mitochondrial membrane has been subject to numerous investigations. For many years it was assumed that the respiratory chain complexes exist as separate units in the fluid lipid bilayer of the inner membrane and interact by random collision (*Hackenbrock et al., 1986*). The discovery of supercomplexes by blue-native polyacrylamide gel electrophoresis (BN-PAGE) after solubilization with mild detergents (*Schägger and Pfeiffer, 2000*) gave rise to the plasticity model, which postulates that respiratory complexes can exist both free in the membrane and as larger supramolecular entities (*Acin-Perez et al., 2008*; *D'Aurelio et al., 2006*). Since then, several stoichiometric supercomplexes have been identified, amongst which the respirasome (supercomplex $I_1III_2IV_1$) is the most prominent. The respirasome contains all components required to transfer electrons from NADH to molecular oxygen (*Schägger and Pfeiffer, 2000*).

The possible functional and structural roles of supercomplexes have been hotly debated. At the functional level, advantages due to partitioning of the quinol pool and substrate channeling have been postulated and are supported by several independent studies (*Bianchi et al., 2003*; *Lapuente-Brun et al., 2013*). Other results are inconsistent with a partitioning of the quinol pool (*Blaza et al., 2014*). A reduction in the level of reactive oxygen species (ROS) that are generated as side products of electron transfer reactions in the respiratory chain has also been suggested as a possible role (*Maranzana et al., 2013*; *Panov et al., 2006*; *Seelert et al., 2009*). From the point of view of protein structure, supercomplexes have been proposed to confer stability to complex I or assist in its assembly (*Marques et al., 2007*; *Schägger et al., 2004*). In line with this, a model for the generation of supercomplexes was proposed, where the assembly of catalytic subunits of the complex I NADH: dehydrogenase module occurs at a late stage to activate the supercomplexes (*Moreno-Lastres et al., 2012*). However, recent complexome profiling studies failed to detect supercomplexes containing immature complex I, suggesting that the respirasome forms by association of fully assembled component complexes (*Guerrero-Castillo et al., 2016*).

Several mitochondrial disorders are associated with impaired respirasome formation. Genetic mutations that impair the assembly of complex III result in a loss of complex I and combined complex III/I defects (*Acin-Perez et al., 2004*; *Bruno et al., 2003*; *Lamantea et al., 2002*; *Schägger et al., 2004*). Complex IV deficiencies associated with a reduction of complex I levels in mouse and human cells have also been reported (*D'Aurelio et al., 2006*; *Diaz et al., 2006*; *Vempati et al., 2009*). The secondary loss of complex I upon impaired expression of complexes III and IV has been taken to mean that complex I stability depends on physical interaction with other complexes in the respiratory chain, since pharmacological inhibition was not sufficient to reduce complex I levels to the same extent (*Acin-Perez et al., 2004*; *Diaz et al., 2006*). Recent studies suggest however that low levels of cytochrome $bc_1$ and cytochrome *c* oxidase (as well as cytochrome *c*) result in an accumulation of reduced quinone, which would trigger reverse electron transfer (RET) and generation of superoxide by complex I. This could result in oxidative damage and complex I degradation (*Guaras et al., 2016*). When ROS production by RET was inhibited, complex I levels were restored (*Guaras et al., 2016*).

Structures of the isolated respirasome at estimated resolutions of 20–30 Å have been obtained by negative-stain electron microscopy (EM) (*Schäfer et al., 2007*), single-particle cryo-EM (*Althoff et al., 2011*) and electron cryo-tomography (*Dudkina et al., 2011*). However, the map resolution was insufficient to fit the component complexes precisely, or to detect new functionally relevant features. Moreover, the recently published high-resolution cryo-EM structures of bovine complex I (*Vinothkumar et al., 2014*; *Zhu et al., 2016*) contribute to a comprehensive view of the respirasome. Here, we report a cryo-EM map of the supercomplex $I_1III_2IV_1$ from bovine heart mitochondria at 9 Å resolution. We show specific protein-protein contacts between the three respiratory chain complexes within the respirasome. Our structure reveals a functionally asymmetric complex III, in which one monomer preferentially catalyzes the reduction of cytochrome *c* in a defined supramolecular organization of the electron transport chain.

Two cryo-EM studies of the respirasome from ovine and porcine heart mitochondria have been published since this manuscript was submitted (*Gu et al., 2016*; *Letts et al., 2016b*). Both are at significantly higher resolution than our structure, but neither of them observe the functional asymmetry

of complex III, which we regard to be the most important functional insight from the respirasome structure. A detailed comparison has been added to the discussion.

## Results

### Isolation of mitochondrial supercomplexes solubilized with PCC-a-M

Since the discovery of mitochondrial supercomplexes (*Schägger and Pfeiffer, 2000*), digitonin has been the detergent of choice for their solubilization and characterization. Digitonin is very suitable for the isolation of supercomplexes $I_1III_2IV_1$ and $I_1III_2$ (*Figure 1A*), and density gradient centrifugation has been the preferred method for their purification. However, to separate the two digitonin-solubilized supercomplexes on a preparative scale by this method proved to be challenging, due to their small difference in density.

PCC-a-M (trans-4-(trans-4'-propylcyclohexyl)cyclo-hexyl-α-D-maltoside) is a recently developed mild, non-ionic detergent (*Hovers et al., 2011*). In an initial screen, we identified a narrow concentration range in which PCC-a-M solubilizes mitochondria efficiently, while preserving the interactions between complexes I, III and IV (*Figure 1A*). Inspection by 2D BN/BN-PAGE and in-gel activity assays (*Figure 1—figure supplement 1A,B*) indicated that this procedure yielded a sample that was active and highly enriched in supercomplex $I_1III_2IV_1$. The smaller supercomplex $I_1III_2$ was not detected on the gel.

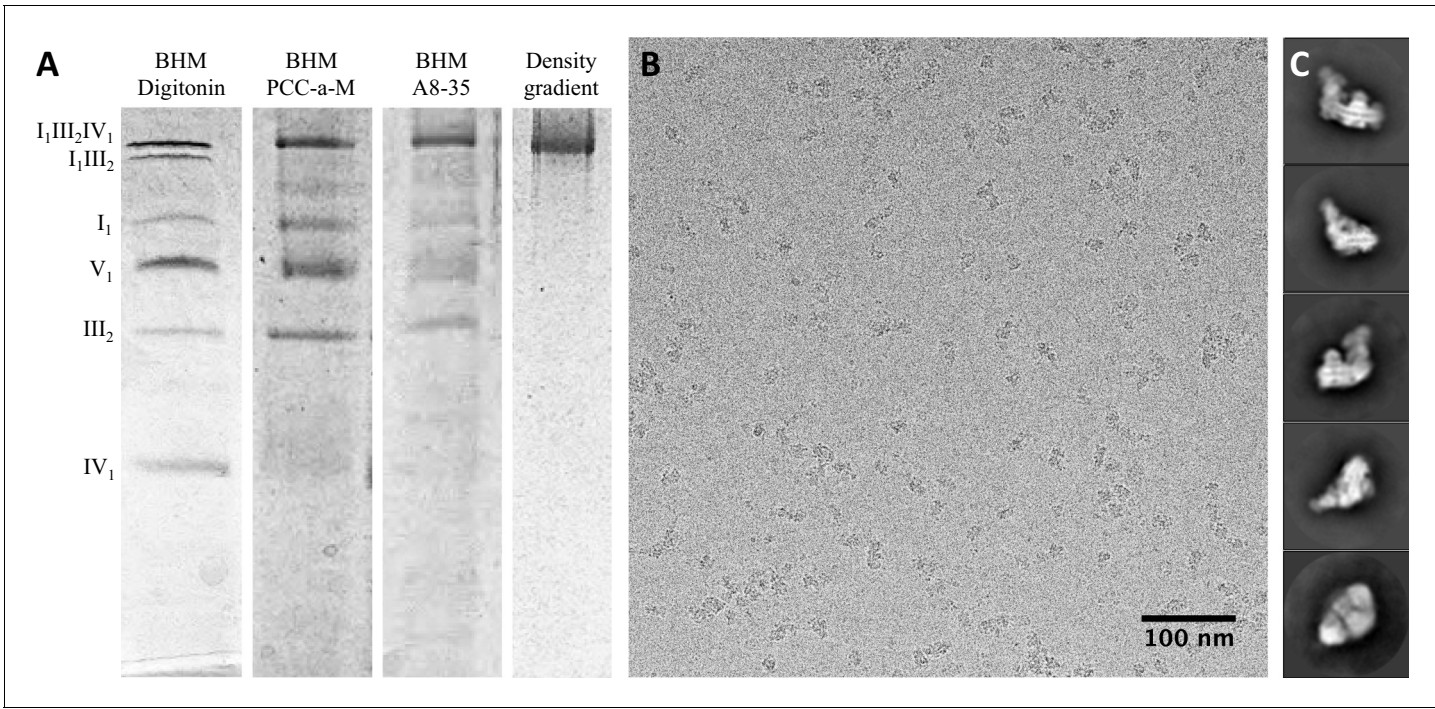

**Figure 1.** Isolation and single-particle cryo-EM of the bovine supercomplex $I_1III_2IV_1$. (**A**) BN-PAGE of bovine heart mitochondria (BHM) solubilized with digitonin (lane 1) or PCC-a-M (lane 2); PCC-a-M-solubilized complex after exchange to A8-35 (lane 3); density gradient fraction of supercomplex $I_1III_2IV_1$ in A8-35 (lane 4). (**B**) Electron micrograph of bovine respirasomes in vitrified buffer, recorded with a Falcon III direct detector in integrating mode on a FEI Tecnai Polara electron microscope operating at 300 kV. (**C**) 2D class averages produced by reference-free 2D classification of 156,536 particles in RELION 1.4.

The following figure supplements are available for figure 1:

**Figure supplement 1.** Assessment of sample quality.

**Figure supplement 2.** 3D classification and refinement.

For further purification, the supercomplex was transferred from PCC-a-M to amphipol A8-35, in which density gradient centrifugation has been shown to work well (*Althoff et al., 2011*). Gradient centrifugation in amphipol A8-35 produced essentially pure supercomplex $I_1III_2IV_1$ (*Figure 1A*). In particular, supercomplex $I_1III_2$ was not detectable. Image analysis of negatively stained samples confirmed that supercomplex $I_1III_2IV_1$ purified from membranes solubilized with PCC-a-M was pure, whereas with digitonin, 42% of the particles had lost complex IV (*Figure 1—figure supplement 1C*).

## Structure determination by single-particle cryo-EM

Cryo-EM grids of supercomplex $I_1III_2IV_1$ in amphipol A8-35 were prepared immediately after purification and electron micrographs were recorded (*Figure 1B*). A total of 156,519 particles was picked manually and classified in RELION 1.4 (*Figure 1C*). Class averages with recognizable views of the supercomplex were selected, and the resulting 137,606 particles were refined, using an earlier low-resolution cryo-EM map (*Althoff et al., 2011*) low-pass filtered to 60 Å as initial reference. The resulting overall average map had a nominal resolution of 8.1 Å (*Figure 1—figure supplement 2*). The map definition varied considerably across the structure, suggesting that it might be an average of several different assemblies or conformations. Multiple rounds of 3D classification indeed revealed a high degree of compositional and conformational heterogeneity. The most homogeneous class of 17,094 particles (class 1; 11% of the total) yielded a structure of the respirasome at a final resolution of 9.1 Å. Notwithstanding its slightly lower nominal resolution as a consequence of the smaller number of particles, the features of this map are very much clearer than in the global average. A second structure of supercomplex $I_1III_2IV_1$ with distinctly different features was resolved to 10.4 Å (class 2; 6% of the total). Lastly, a third class with 12,042 particles lacking complex IV (class 3; 8% of the total) was refined to 9.9 Å (*Figure 1—figure supplement 2*).

## Defined protein-protein contacts

The three component complexes $I_1$, $III_2$ and $IV_1$ of the respirasome are well-resolved in class 1 (*Figure 2A*) and class 2. At an estimated local resolution of 8.6 Å (*Figure 2—figure supplement 1*), the transmembrane region of class 1 indicates clear densities for most of the 132 membrane-spanning α–helices in the supercomplex (*Figure 2B*). Atomic models for bovine complex I (*Vinothkumar et al., 2014*), III (*Iwata et al., 1998*) and IV (*Tsukihara et al., 1996*) were docked into the 3D maps. The fit was excellent for all three complexes in all map regions. A good fit of all three complexes was also obtained for class 2, whereas class 3 lacked the density for complex IV and thus corresponded to supercomplex $I_1III_2$.

The presence of supercomplex $I_1III_2$ and of free complex I (15% of the initial data set; *Figure 1—figure supplement 2*) in the cryo-EM sample was surprising, since biochemical analysis and negative-stain EM had both shown that the amphipol-solubilized complex was stable and free of these assemblies (*Figure 1—figure supplement 1*). Evidently, supercomplex $I_1III_2IV_1$ partly dissociated on the cryo-EM grid prior to freezing, either as the result of an increase in ionic strength due to evaporation, or of surface forces at the air-water interface.

The class 1 map shows that the three component complexes are in close contact with each other at defined points primarily near the membrane surfaces (*Figure 2B*). With one exception, all inter-complex contacts involve supernumerary subunits. The most extensive interactions occur between complexes I and III. In the matrix, subunit B22 of complex I is in touch with subunit 1 of complex III at a minimum distance of 4 Å. In the membrane region, subunit B14.7 of complex I approaches subunit 8 of cytochrome $bc_1$ near both membrane surfaces to within sidechain contact (*Figure 3*). In the centre of the supercomplex, subunit B14.7 establishes clear contacts with complex III within the hydrophobic interior of the respirasome.

Contacts between complexes I and IV are mediated by subunit ND5 of complex I that approaches complex IV subunit COX7C to within an α–carbon distance of around 10 Å at the membrane surface on the matrix side. ND5 appears to be the only one of the 14 complex I core subunits that is involved directly in supercomplex formation. The interface between complexes III and IV is formed by subunits 1, 9 and 10 of the $bc_1$ complex and COX7A1 of cytochrome $c$ oxidase. Subunit 9 of complex III has previously been shown to be required for stable interaction with cytochrome $c$ oxidase (*Zara et al., 2007*).

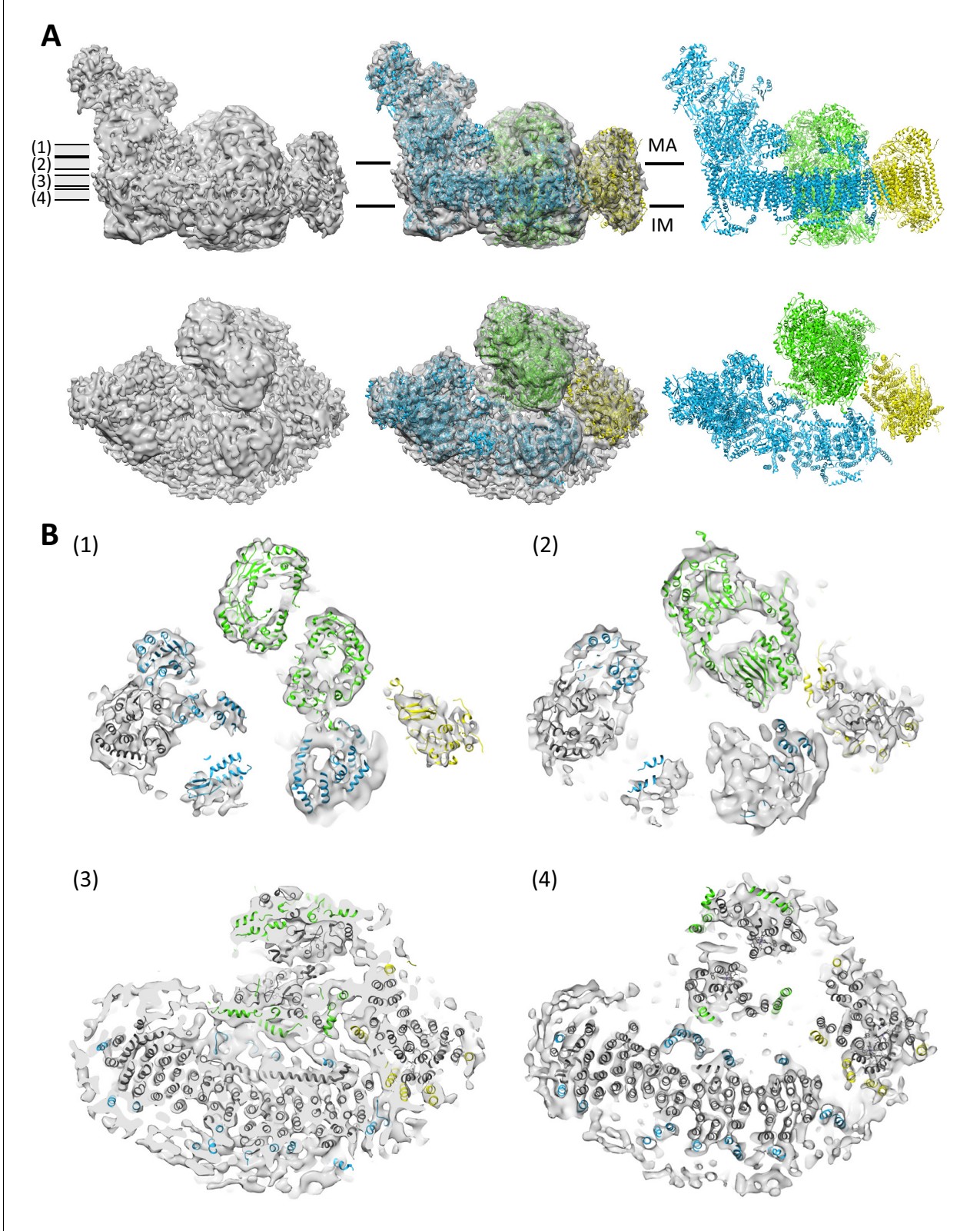

**Figure 2.** Protein-protein contacts in the respirasome. Most contacts are mediated by supernumerary subunits of complexes I, III and IV. (**A**) Side views (upper row) and views from the matrix (bottom row) of class 1 cryo-EM map filtered to 8.6 Å with docked atomic models of complexes I (blue) (*Vinothkumar et al., 2014*), III (green) (*Iwata et al., 1998*) and IV (yellow) (*Tsukihara et al., 1996*). (**B**) Slices through the map at positions shown in (**A**) indicate contact points between the three complexes in the membrane. Core subunits are shown in grey and supernumerary subunits in color.
*Figure 2 continued on next page*

*Figure 2 continued*

The following figure supplement is available for figure 2:

**Figure supplement 1.** Local map resolution.

## Conformational and compositional variability

A significant portion of the particles clustered in class 2, where the mutual arrangement of complexes III and I is different from that in class 1, whereas the interaction of complex I with complex IV is unchanged (*Figure 4A*). Classes 1 and 2 are related by a 25° rigid-body rotation of the whole complex III dimer relative to complex I, around an axis roughly perpendicular to the membrane plane. This rotation breaks the protein-protein contacts of complex III with complexes I and IV observed in class 1. The particles that cluster in class 2 therefore appear to represent a different form of the respirasome. There were no significant new interactions between complex III and complexes I and IV in this class, implying that it may be less stable. In class 2, the matrix arm of complex I has moved away from the center of the supercomplex by a rotation of 3–4° around an axis parallel to the membrane (*Figure 4B*). Structures of active and deactive states of complex I have been recently presented, which differ by a similar movement of the matrix arm (*Zhu et al., 2016*). Class 2 may thus represent a subpopulation of the supercomplex in which complex I is in the deactive state. Alternatively, class 2 may result from rearrangement or destabilization of the supercomplex during purification. The mutual arrangement of complexes I and III in class 3 is the same as in class 1, confirming that the interaction between these two proteins is sufficiently stable to fix them in this specific conformation (*Figure 4A and B*).

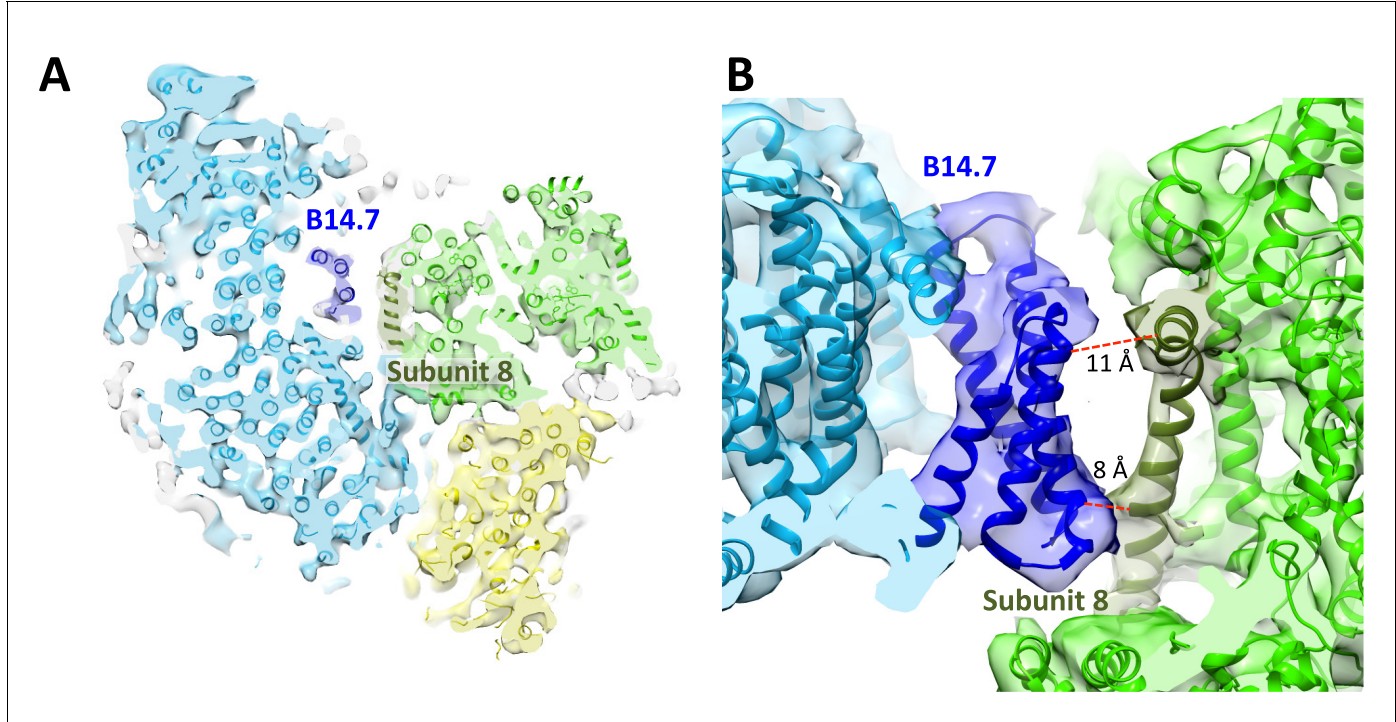

**Figure 3.** Central position of complex I supernumerary subunit B14.7. (**A**) Horizontal slice through 8.6 Å map, with fitted models for complexes I (light blue), III (light green) and IV (yellow). Subunit B14.7 of complex I is dark blue and subunit 8 of complex III is dark green. (**B**) Detailed side view of interface shows close contacts between complex I B14.7 and complex III subunit 8 in the hydrophobic interior of the supercomplex. Distances are between α–carbons in polypeptides of adjacent complexes.

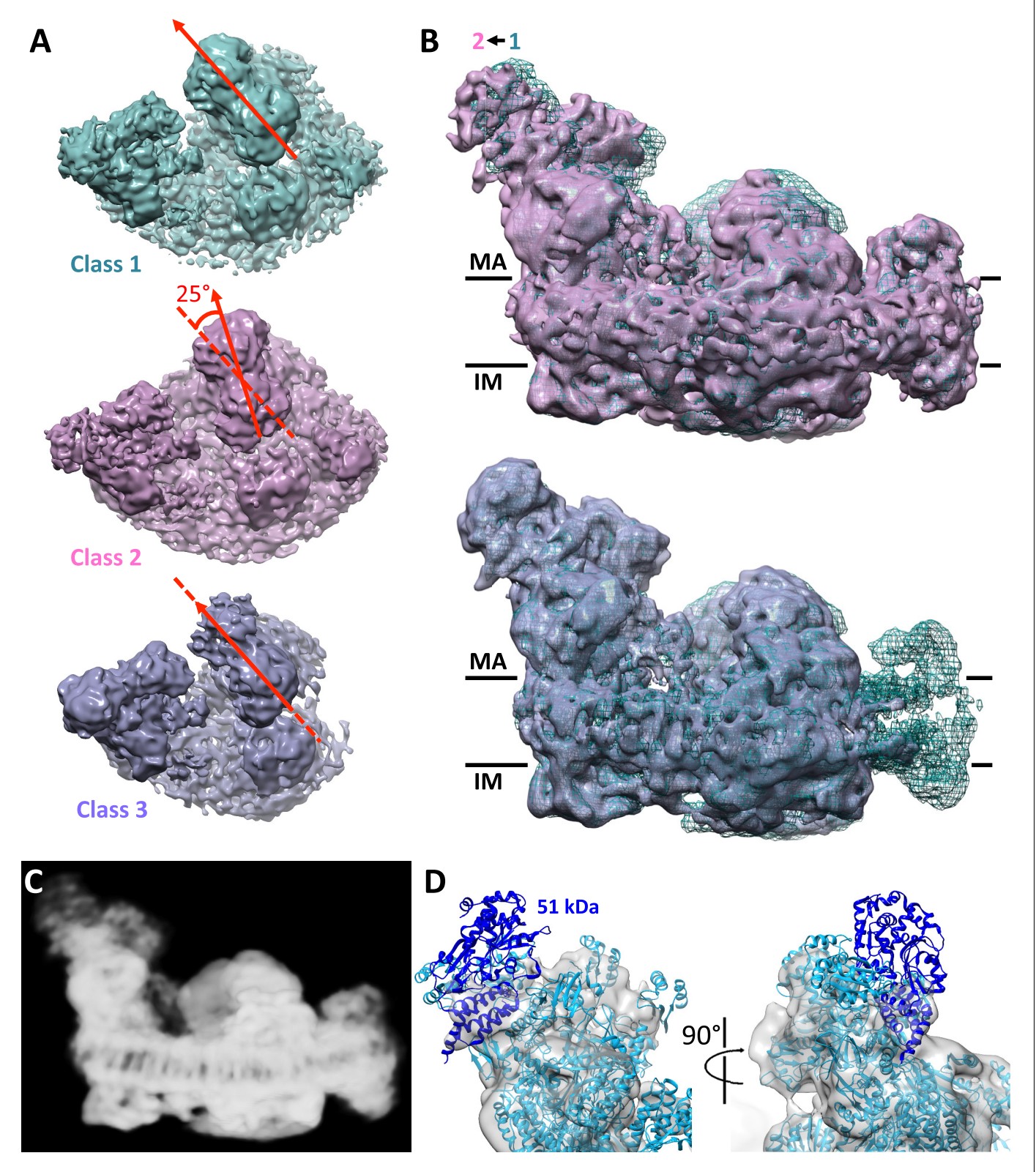

**Figure 4.** Conformational and compositional heterogeneity of the respirasome. (**A**) Top view of density maps of class 1 (dark cyan) and class 2 (pink) of the respirasome and supercomplex $I_1III_2$ (violet). Classes 1 and 2 differ by a 25° rigid-body rotation of complex III in the membrane plane relative to complex I. Complex IV occupies identical positions in both maps. The arrangement of complex I and III in class 3 is similar to that in class 1. (**B**) Side views of classes 2 and 3 overlayed with the class 1 map (dark cyan mesh) indicate flexibility of the complex I matrix arm. In class 2 the matrix arm is

*Figure 4 continued on next page*

*Figure 4 continued*
displaced by 3–4° away from the center of the map. (**C**) Consensus map of the respirasome at 8.1 Å with weak density at the distal part of the matrix
arm. (**D**) Complex I matrix arm in the consensus map with fitted atomic model of complex I with the 51 kDa subunit in dark blue seen from the front
(left) and side (right).

Several of the classes obtained (~30% of the initial dataset) have no density for the NADH:dehydrogenase module of complex I, or even lack the whole matrix arm (*Figure 1—figure supplement 2*). While these classes might reflect a partial loss of this module during purification or cryo-EM grid preparation, we cannot rule out the possibility that they represent biologically relevant assembly, disassembly or recycling intermediates of the respirasome. A recent study of respirasome biogenesis (*Moreno-Lastres et al., 2012*) has suggested that supercomplexes form by association of individual subunits of the respiratory complexes or their intermediate assemblies, rather than from the fully assembled respiratory chain complexes. The incorporation of the NADH:dehydrogenase module of complex I has been presented as the final step in the assembly process (*Moreno-Lastres et al., 2012*). The study reported a significant delay in the integration of the 51 kDa subunit in this module, which is one of the core subunits of the N-catalytic unit of complex I. These observations are in agreement with our 8.1 Å global average (*Figure 4C,D*), in which the density for the 51 kDa subunit is weak and could explain why the NADH:dehydrogenase module is absent in several of the 3D class averages (*Figure 1—figure supplement 2*). However, other studies suggest that supercomplexes form by association of fully assembled component complexes (*Guerrero-Castillo et al., 2016*), which would thus not account for the intermediate structures we observe.

The presence of supercomplex assemblies lacking N module subunits is equally consistent with different turnover rates of complex I subunits. It has been shown that several of the nuclear-encoded subunits in the mature enzyme can be exchanged against newly imported copies, in parallel to the *de novo* assembly of the complex (*Dieteren et al., 2012*; *Lazarou et al., 2007*). This holds true for most subunits of the N-catalytic unit (*Dieteren et al., 2012*; *Lazarou et al., 2007*), which are particularly susceptible to oxidative damage. A direct replacement of these subunits would provide an efficient means of repairing oxidative damage of complex I (*Dieteren et al., 2012*; *Lazarou et al., 2007*). These classes therefore might represent intermediates of the supercomplex, generated during its assembly or for complex I regeneration.

## Functional asymmetry and electron flow in complex III

The most striking and unexpected feature of the respirasome map is that one of the two membrane-extrinsic iron-sulfur domains of the complex III dimer shows clear density (*Figure 5A*). The iron-sulfur domain is the active part of the Rieske protein subunit of the $bc_1$ complex that carries electrons from the $Q_P$ site via its iron-sulfur cluster to heme $c_1$, where they are picked up by the small, soluble cytochrome $c$ protein for transfer to complex IV (*Kühlbrandt, 2015*). Interestingly, the iron-sulfur domain that bridges the gap in the electron transfer path between the hemes of one complex III monomer belongs to the Rieske protein of the opposite monomer. In order to transfer electrons to $c_1$, the iron-sulfur domain has to move on a hinge by 10–15 Å (*Iwata et al., 1998*). Such a movement would make the iron-sulfur domain invisible in a 9 Å map. The hinge movement of the iron-sulfur domain is essential for function since when it is blocked, no electrons are transferred to cytochrome $c_1$ (*Darrouzet et al., 2000*). The fact that one of the two iron-sulfur domains in the supercomplex is resolved means that one of the two complex III monomers is inactive. The inactive monomer is the one in contact with the well-defined iron-sulfur domain. This monomer is exposed to the lipid bilayer on most of its periphery, except for the hydrophobic surface area that is involved in complex III dimer formation. The active monomer near the disordered iron-sulfur domain is surrounded by complex I, complex IV and the inactive complex III monomer (*Figure 5A*).

It is generally assumed that both monomers of the complex III dimer participate equally in mitochondrial electron transport. However, for the bacterial cytochrome $bc_1$ complex, which is a simpler version of mitochondrial complex III and likewise dimeric, it has been shown that there is no difference in the electron transport activity of the $bc_1$ complex if one of its monomers is permanently inactivated by mutagenesis (*Castellani et al., 2010*). Therefore, in bacteria only one half of the $bc_1$ dimer is required for electron transfer, and this most likely holds true for the mitochondrial complex

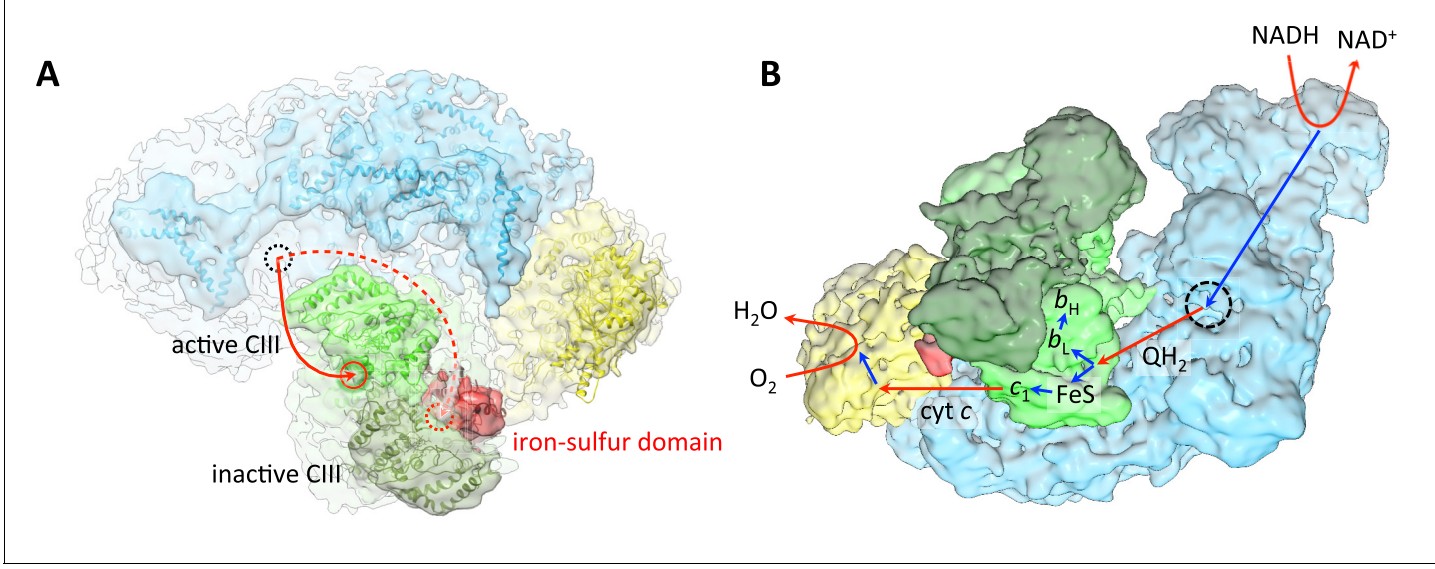

**Figure 5.** Substrate and electron flow in the respirasome. (**A**) Respirasome map with fitted atomic models (blue, complex I; green, complex III; yellow, complex IV) seen from the crista lumen. Only one of the two Rieske iron-sulfur domains of complex III is resolved (red), indicating that the complex III monomer associated with it is inactive. The active complex III monomer is light green, the inactive monomer is dark green. The red circles indicate the position of the ubiquinol binding sites of complex III in the intermembrane side of the membrane. The dashed black circle indicates the position of the quinol binding site of complex I on the matrix side. (**B**) Oblique view from the lumenal side of the surface-rendered respirasome map, with routes of electron and substrate transfer. Red and blue straight arrows indicate substrate and electron transfer, respectively. Respiratory chain reactions are indicated by curved arrows. Electrons from NADH pass through complex I and reduce quinone to quinol in the quinol binding site (dashed black circle). Reduced quinol diffuses in the membrane, binding preferentially to the $Q_P$ site of the central, active monomer of complex III. Quinol is oxidized to quinone, transferring one electron to heme $b_L$ for quinone reduction at the $Q_N$ site, and another electron to the iron-sulfur cluster of the Rieske protein in the $b$ position. The iron-sulfur domain moves from the $b$ to the $c_1$ position and reduces heme $c_1$. Heme $c_1$ reduces cytochrome $c$, which diffuses to complex IV where it donates electrons for reduction of $O_2$ to $H_2O$.

as well. In the bacterial complex, the choice of the active monomer is thought to be random (*Covian and Trumpower, 2008*), whereas in the respirasome, the active monomer is clearly the one in the centre of the supercomplex. At present we do not know what constrains the movement of one of the two iron sulfur domains in the supercomplex. Either electrostatic or steric effects in the asymmetrical environment of the respirasome may be responsible. In an unconstrained complex III dimer, both iron-sulfur domains should be mobile, and hence invisible in the map.

## Discussion

### Electron and substrate transfer

Under normal turnover conditions, the two monomers of complex III are not active simultaneously (*Castellani et al., 2010*). A model for electron transfer within complex III has been proposed, in which the activation of either monomer is random and both monomers switch between active and inactive states (*Covian and Trumpower, 2008*). This model implies that the two branches for electron transfer in complex III are equivalent and, in particular, both iron-sulfur domains will move between their $b$ and $c_1$ positions (*Iwata et al., 1998*) for reduction of heme $c_1$ and, subsequently, cytochrome $c$.

In the various crystal structures of complex III, the iron-sulfur domains occupy different positions, either as a result of crystal contacts or, more significantly, in response to the occupancy of the $Q_P$ site (*Berry et al., 2013*). In single particle cryo-EM conformational changes are not subject to such contacts. Therefore, in the absence of inhibitors, and in light of the known mechanism of electron transfer, a cryo-EM map of complex III should reveal two equally unresolved iron-sulfur domains, as a result of averaging particles in different states. In our map, only one iron-sulfur domain is

unresolved, and hence active. The observed asymmetry is thus incompatible with an alternating activation of the two complex III monomers and implies that each monomer has a different function. Our data suggest that in the respirasome, the central monomer of complex III preferentially catalyzes quinol oxidation. As electrons can equilibrate rapidly between the two $b_L$ hemes (*Castellani et al., 2010*), quinone reduction may occur in either monomer, although it is possible that the distal monomer has no function, except as a scaffold.

In the respirasome, the quinol oxidation site of the active complex III monomer faces the site on complex I where the reduced quinol is released. The shortest distance between the two sites is about 11 nm (*Figure 5A*). The quinol oxidation site of the inactive monomer is on the far side of the dimer, at a distance of at least 18 nm from the complex I site, assuming that the quinol cannot pass through the complex III dimer. A quinol diffusing through the lipid bilayer from complex I to complex III would thus encounter the active monomer first. The linear arrangement of complex I, the active complex III monomer and complex IV in the respirasome seems to be particularly favourable for efficient substrate and electron transfer. Significantly, only one of the complex III monomers is well-placed to accept reduced quinol from complex I, which may explain why only this monomer is active. Substrate channeling within the respirasome has been extensively debated. Results inconsistent with the existence of distinct quinone pools and substrate channeling have been reported (*Blaza et al., 2014*), but substrate channeling between complexes I and III is supported by most studies of metabolic flux control (*Bianchi et al., 2003*; *Lapuente-Brun et al., 2013*) and in recent reviews discussing the results of Blaza et al (*Enríquez, 2016*; *Lenaz et al., 2016*). The asymmetry observed for complex III in our supercomplex structure is in agreement with these results and provides further support for electron transfer by substrate channeling in the respirasome.

While it is clear that all functional complex III in mitochondria is dimeric, it is not known at this stage what proportion of the complex III dimers is incorporated into respirasomes. The functional asymmetry of complex III and its iron-sulfur domains in the supercomplex provides strong evidence for a spatially organized flow of electrons and substrates in the respiratory chain from complex I to complex IV (*Figure 5B*). Turning off one of two electron transfer paths through complex III that are, in principle, equally likely might also offer a kinetic advantage in the electron transfer to complex IV via cytochrome *c*. The cytochrome *c* binding site on the active $bc_1$ monomer is about 10 Å closer to complex IV than that of the inactive monomer. Complex IV needs four electrons, and hence four cytochromes *c,* to produce one molecule of water. The proximity of the binding sites, which would minimize the time for each of the four cytochrome *c* transfer steps, may be an important advantage.

Our observation that in the respirasome only one of two possible electron transfer branches is active is reminiscent of the two near-symmetrical electron transfer chains in photosystem II, one of which has been turned off in the course of evolution to avoid wasteful electron transfer to two identical quinol electron acceptors (*Nelson and Yocum, 2006*). In some ways, oxygen reduction in complex IV is the reverse of the water oxidation reaction in photosystem II, and it might be equally important in both cases to control the flow of four electrons into, or away from, the reaction site by turning off one of two possible transfer paths.

## Supercomplex assembly, complex I stability and human health

It is widely known that dysfunctional electron transfer complexes of the mitochondrial respiratory chain cause severe and, at present, incurable genetic disorders. Many of these conditions can be traced to point mutations in subunits of mitochondrial complex I (*Rodenburg, 2016*) or III (*Meunier et al., 2013*). Many mutations result in reduced levels or activities of either complex I or III in mitochondrial membranes. However, point mutations (*Bruno et al., 2003*) or truncation (*Lamantea et al., 2002*) of the mitochondrially encoded cytochrome *b* subunit of complex III that cause progressive exercise intolerance and lactic acidosis in human patients were found to not only reduce the levels of complex III but also that of complex I. Biochemical studies of similar mutants established that the depletion of complex III levels in the membrane results in a secondary loss of complex I (*Acin-Perez et al., 2004*; *Schägger et al., 2004*). More recently, stable expression of complex I in the absence of complex III has been demonstrated through direct or indirect inhibition of ROS production by RET (*Guaras et al., 2016*). Inhibition of cytochrome $bc_1$ activity however was insufficient to induce complete degradation of complex I (*Acin-Perez et al., 2004*; *Guaras et al., 2016*), suggesting that physical interaction between the complexes has an at least partially stabilizing effect. These findings point to a central role of the respirasome in complex I stability.

The nuclear-encoded, supernumerary complex I subunit B14.7 (*Figure 3*) evidently has a major part in the formation and stability of the respirasome. A point mutation in this subunit is associated with complex I deficiency in patients with fatal infantile lactic acidemia (*Berger et al., 2008*), and a disruption of the gene of the homologous protein in the fungus *Neurospora crassa* results in incomplete assembly of complex I (*Nehls et al., 1992*). At present it is not known whether these mutations affect the assembly of the respirasome, but on the basis of our structure it seems more than likely.

## Mammalian respirasome structures

While this manuscript was under review, two other cryo-EM structures of mammalian respirasomes were published (*Gu et al., 2016*; *Letts et al., 2016b*). These studies used mitochondria from porcine or ovine, rather than bovine, heart. The ovine complex I has been described as being particularly stable and suitable for structural studies (*Letts et al., 2016a*) and the same most likely holds true for supercomplexes. At resolutions of 5–6 Å, both structures corroborate most of the protein-protein contacts we observe in the bovine complex. Several additional subunits were found to participate in the stabilization of the supercomplex, which were not clearly visible at our map resolution. A detailed comparison is possible only for the ovine complex, since neither the map nor the coordinates of the porcine supercomplex have been released.

Of the two new studies, only one (*Letts et al., 2016b*) found a second significant respirasome conformation at an overall resolution of 6.7 Å. The two states are referred to as the tight and loose conformation. The tight conformation of the ovine complex is essentially identical to class 1 of the bovine respirasome, but the arrangement of component complexes in the loose state differs considerably from our class 2. In the bovine respirasome, complex III rotates by 25° relative to complex I, while the position of complex IV remains unchanged. In contrast, in the loose ovine respirasome, complex IV swings away from complex III. Complex III rotates in the same direction as in our class 2, but to a lesser extent. In both the bovine and the ovine respirasome, interactions between complex III and IV are disrupted in these minor classes. In the case of the ovine complex, the proportion of particles in the loose state was found to increase with time (*Letts et al., 2016b*). This suggests that the loose state of the ovine respirasome and our class 2 of the bovine complex result from incipient denaturation during and after purification. The differences observed between these two classes most likely reflect different degrees of respirasome instability, which may be species-specific, or due to differences in supercomplex purification.

Surprisingly, the two iron-sulfur domains in the complex III dimer are both resolved in the porcine and ovine respirasome, whereas one of them is disordered in our structure of the bovine complex. Therefore neither of the two other structures shows a mobile and therefore catalytically active Rieske protein, as we observe in the bovine respirasome. Letts et al propose that due to the close proximity between complex IV and the Rieske protein of the outer monomer of complex III, the iron-sulfur domain of the outer complex III monomer cannot undergo the conformational changes required for electron transfer. This is consistent with our findings, although neither structure shows any direct protein-protein contacts that would inhibit domain movements. Letts et al conclude that quinone oxidation is likely to take place in the central complex III monomer. This agrees with our model, which is however based on firm experimental evidence of an unresolved, and hence mobile, iron-sulfur domain. Letts et al also propose that quinone reduction in complex III should be catalyzed by the distal monomer; however the respirasome structures provide no evidence for this. Further studies should clarify the exact transfer path of electrons and quinol substrates through the mammalian respirasome.

## Conclusions

The 9 Å structure of the mammalian mitochondrial supercomplex shows a well-defined arrangement of the respiratory chain complexes I, III and IV from the inner mitochondrial membrane. The ordered structure of one of the two iron-sulfur domains in the complex III dimer indicates that, contrary to expectation, one of the two electron transfer pathways in complex III is always active, whereas the other monomer is inactive. The active complex III monomer is in a pivotal position between the quinol binding site of complex I and the cytochrome *c* binding site of complex IV, and hence for directed electron flow through the respirasome. Far from being a random collection of close-packed respiratory chain complexes, the mitochondrial respirasome has not only a clear role for the long-

term stability of its component complexes, in particular complex I, but also in optimizing electron flow in cellular respiration.

## Materials and methods

### Isolation and purification of supercomplexes from bovine heart mitochondria

Bovine heart mitochondria were prepared by differential centrifugation as described (*Krause et al., 2005*). Mitochondria were solubilized with 1% (w/v) digitonin (Calbiochem) at a detergent-to-protein ratio of 28:1 (*Schäfer et al., 2006*) or with 0.11% (w/v) PCC-a-M (Glycon) at a detergent-to-protein ratio of 1:1 by incubation for 1 hr at 4°C. Detergent exchange to amphipol A8-35 and protein purification were performed as described before (*Althoff et al., 2011*).

### BN-PAGE and in-gel activity assays

Proteins were separated on 3–12% polyacrylamide linear gradient gels by BN-PAGE and 2D BN/BN-PAGE (*Wittig et al., 2006*). Functionally active supercomplexes were detected by in-gel activity assays for NADH:dehydrogenase and cytochrome *c* oxidase (*Grad and Lemire, 2006*; *Kuonen et al., 1986*). Briefly, for complex I, gels were incubated in buffer containing 100 mM Tris, pH 7.4, 0.5 mM NBT (*p*-nitrotetrazolium blue) and 100 µM β–NADH for 1 hr. For complex IV, gels were incubated for 10–12 hr in 50 mM sodium phosphate buffer at pH 7.2, 0.5 mg/ml DAB (3,3'diaminobenzidine tetrahydrochloride), 1 mg/ml equine heart cytochrome *c* and 20 U/ml bovine liver catalase.

### Data collection

3 µl of a respirasome sample were applied to freshly glow discharged Quantifoil R2/2 holey carbon grids (Quantifoil Micro Tools, Germany) that had been pretreated in chloroform for 1–2 hr. The grids were blotted for 10 s at 90% humidity and 10°C in an FEI Vitrobot plunge freezer. Immediately before freezing, 0.6 µl of 1 M KCl were added to avoid protein aggregation. Cryo-EM images were collected on a FEI Tecnai Polara operating at 300 kV aligned as previously described (*Mills et al., 2013*). The microscope was equipped with a Falcon III direct electron detector. Images were recorded manually at a nominal magnification of 59,000x yielding a pixel size at the specimen of 1.77 Å. The camera system recorded 32 frames/s. Videos were collected for 1.5 s with a total of 46 frames with a calibrated dose of about 1.5 e⁻/Å² per frame, at defocus values between −1.3 and −4.3 µm.

### Image processing

A set of 3592 micrographs was collected. Whole-image drift correction of each movie was performed using the algorithm developed by Li (*Li et al., 2013*). Particles were picked manually using EMAN Boxer (*Ludtke et al., 1999*), and the micrograph-based CTF was determined using CTFFIND3 (*Mindell and Grigorieff, 2003*) in the RELION 1.4 workflow (*Scheres, 2012*). The initial dataset contained 156,519 particle images (288 pixels x 288 pixels). Particles were subjected to two-dimensional reference-free classification in RELION 1.4 (*Scheres, 2012*). Visual selection of particle classes with interpretable features resulted in a dataset of 137,606 particle images for the first 3D consensus refinement. The earlier low-resolution respirasome map (*Althoff et al., 2011*) was low-pass filtered to 60 Å and used as an initial model for the 3D refinement in RELION 1.4. Individual frames were B-factor weighted and movements of individual particles were reversed by movie frame correction in RELION 1.4 (*Scheres, 2014*). The resulting dataset of polished particles was used for 3D classification and the best 3D classes were selected for further processing. UCSF Chimera (*Pettersen et al., 2004*) was used for visualization of cryo-EM maps and docking of atomic models. Figures were drawn with UCSF Chimera.

### Data deposition

The cryo-EM maps were deposited in the Electron Microscopy Data Bank with accession code EMD-4107, EMD-4108 and EM-4109 for the cryo-EM maps of class 1, 2 and 3, respectively and the

structure coordinates of the complexes fitted to class 1 were deposited in the Protein Data Bank with accession number 5LUF.

## Acknowledgements

We thank Matteo Allegretti for help with cryo-EM data acquisition and image processing, and Natalie Bärland for help with particle picking. This work was funded by the Max Planck Society and the Cluster of Excellence Frankfurt 'Macromolecular Complexes' (DFG Project EXC 115).

## Additional information

### Competing interests

WK: Reviewing editor, *eLife*. The other authors declare that no competing interests exist.

### Funding

| Funder | Grant reference number | Author |
| --- | --- | --- |
| Max-Planck-Gesellschaft | | Werner Kühlbrandt |
| Cluster of Excellence Frankfurt | DFG Project EXC 115 | Werner Kühlbrandt |

The funders had no role in study design, data collection and interpretation, or the decision to submit the work for publication.

### Author contributions

JSS, Wrote the article, Acquisition of data, Analysis and interpretation of data; DJM, Supported the cryo-EM work, Drafting or revising the article; JV, Conception and design, Analysis and interpretation of data, Drafting or revising the article; WK, Wrote the article, Conception and design, Analysis and interpretation of data

### Author ORCIDs

Janet Vonck, http://orcid.org/0000-0001-5659-8863
Werner Kühlbrandt, http://orcid.org/0000-0002-2013-4810

## Additional files

### Major datasets

The following datasets were generated:

| Author(s) | Year | Dataset title | Dataset URL | Database, license, and accessibility information |
| --- | --- | --- | --- | --- |
| Sousa JS, Mills DJ, Vonck J, Kuehlbrandt W | 2016 | Cryo-EM map of bovine respirasome | http://www.ebi.ac.uk/pdbe/entry/emdb/EMD-4107 | Publicly available at the EBI Protein Data Bank (accession no: EMD-4107) |
| Sousa JS, Mills DJ, Vonck J, Kuehlbrandt W | 2016 | Cryo-EM of bovine respirasome | http://www.ebi.ac.uk/pdbe/entry/emdb/EMD-4108 | Publicly available at th EBI Protein Data Bank (accession no: EMD-4108) |
| Sousa JS, Mills DJ, Vonck J, Kuehlbrandt W | 2016 | Cryo-EM of bovine respirasome | http://www.ebi.ac.uk/pdbe/entry/emdb/EMD-4109 | Publicly available at the EBI Protein Data Bank (accession no: EMD-4109) |
| Sousa JS, Mills DJ, Vonck J, Kuehlbrandt W | 2016 | cryo-EM of bovine respirasome | http://www.rcsb.org/pdb/explore.do?structureId=5LUF | Publicly available at the RCSB Protein Data Bank (accession no: 5LUF) |

The following previously published datasets were used:

| Author(s) | Year | Dataset title | Dataset URL | Database, license, and accessibility information |
|---|---|---|---|---|
| Vinothkumar KR, Zhu J, Hirst J | 2014 | Electron cryo-microscopy of bovine Complex I | http://www.rcsb.org/pdb/explore/explore.do?structureId=4UQ8 | Publicly available at the EBI Protein Data Bank (accession no: 4UQ8) |
| Iwata S, Lee JW, Okada K, Lee JK, Iwata M, Rasmussen B, Link TA, Ramaswamy S, Jap BK | 1999 | CYTOCHROME BC1 COMPLEX FROM BOVINE | http://www.rcsb.org/pdb/explore/explore.do?structureId=1BGY | Publicly available at the RCSB Protein Data Bank (accession no: 1BGY) |
| Tsukihara T, Aoyama H, Yamashita E, Tomizaki T, Yamaguchi H, Shinzawa-Itoh K, Nakashima R, Yaono R, Yoshikawa S | 1996 | STRUCTURE OF BOVINE HEART CYTOCHROME C OXIDASE AT THE FULLY OXIDIZED STATE | http://www.rcsb.org/pdb/explore/explore.do?structureId=1OCC | Publicly available at the RCSB Protein Data Bank (accession no: 1OCC) |

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
