## [Decision Letter]

Thank you for submitting your article "Functional asymmetry and electron flow in the bovine respirasome" for consideration by *eLife*. Your article has been favorably evaluated by John Kuriyan as the Senior Editor and three reviewers, one of whom, Stephen C. Harrison (Reviewer #1), is a member of our Board of Reviewing Editors.

The reviewers have discussed the reviews with one another and the Reviewing Editor has drafted this decision to help you prepare a revised submission.

Summary and specific request:

This manuscript reports the structure of the bovine respirasome at 9 Å resolution. Its principal conclusion concerns a functional asymmetry: one of the iron-sulfur clusters is docked in such a way that electrons cannot be transferred, because the distance to the partner is too great. The authors conclude that one of the two is inactive. The structure is well determined at the stated resolution, but two structures at much higher resolution appeared after this manuscript was submitted. The reviewers request a revised manuscript that places the current conclusions in the context of those reports. In particular, the authors should address the likelihood of a different interpretation – some sort of alternating access model in which only one subunit can access the transfer site at a time, while the other docks in the observed, autoinhibited position. That is, the statement that the resolved Rieske center is immobile and therefore not functional is one possibility, but just because it has a fixed orientation in this structure of the inactive respirasome does not mean it cannot move in the active respirasome.

The following detailed comments may help with the revision.

Reviewer #2:

It is strange to show an FSC with phases randomized beyond 20 Å. Use of a phase randomized FSC to correct for effects from masking of maps is excellent, but the corrected FSC should be shown, not the phase randomized FSC.

Reviewer #3:

Introduction:

First paragraph: The entrance of electrons in the mitochondrial electron transfer chain is branched, therefore CI is the largest complex, but cannot be called the first, because other enzymes that deliver electrons to CoQ are equally first in their branch. Please correct.

First paragraph: It should be made clear that mammalian CI comprises 44 different subunits, one being duplicated. Quoting Carrol et al. 2002 could be misleading, since in this reference it was considered that CI comprises 45 different subunits. Latter it was demonstrated and generally accepted that NDUFA4 is not a CI subunit. It is advisable to clarify and to use the correct references.

First paragraph: The number of mammalian CIV subunits has been proposed to be 14.

Third paragraph: The issue of CoQ partitioning is still hotly debated. The authors correctly mention that Blaza et al. 2014 disagree with this idea but forget to mention that those authors' experimental setup was rigorously contested in two recent reports (Enriquez 2016. Annu. Rev. Physiol. 78, 533-561 & Lenaz et al., 2016. Biochim Biophys Acta 1857, 991-1000). This is relevant because the very interesting proposal by Sousa et al. for the mechanism of electron transfer within the respirasome agrees with the interpretation of Enriquez and Lenaz rather than that of Blaza.

Third paragraph: The best experimental support for the role of supercomplexes in minimizing ROS production was probably in the paper by Maranzana (Maranzana et al. 2013. Antioxid Redox Signal 19, 1469-1480), which is not quoted.

Third paragraph: The role of supercomplexes in stabilizing CI is noted in the Introduction, but in a way that may not reflect the current view of this question. The historical origin of this proposal is as follows: in 2004 two groups independently proposed that CI stability was compromised in the absence of CIII (Acin-Perez et al., 2004 & Schägger et al. 2004). Latter, the group of Carlos Moraes demonstrated that ablation of CIV or cyt *c*, also compromised the stability of CI (Diaz et al. 2006- Mol Cell Biol 26, 4872-4881 & Vempati, et al. 2009. J. Biol. Chem. 284, 4383-4391). More recently it was demonstrated that CI can be assembled functionally in the absence of CIII, CIV or cyt *c* by recovering the re-oxidation of CoQ (Guarás, 2016. Cell Reports 15, 197-209). This series of reports led to the conclusion that assembly of the individual respiratory complexes can proceed independently and requires neither the presence of other complexes nor interaction with them. In line with this conclusion, kinetic analysis showed that the formation of complexes happened before the formation of supercomplexes (Acín-Peréz et al. 2008. Mol. Cell 32, 529-539). This result does not preclude, however, that incomplete CI or CIII could interact with other complexes and form supercomplexes, as reported by several groups (Marques, I., 2007. Eukaryotic Cell 6, 2391-2405). In 2012 it was proposed that the assembly of the NADH-DH module of CI require that formation of a non-functional respirasome between partially assembled CI with CIII and CIV (Moreno-Lastres et al. 2012. Cell Metab.15, 324-335). This proposal contradicted that of (Acín-Peréz et al. 2008. Mol. Cell 32, 529-539). A very recent reevaluation concluded that CI, CIII and CIV are fully assembled before they associate into supercomplexes, refuting the proposal by Moreno-Lastres (Guerrero-Castillo S, Baertling F, Kownatzki D., Wessels H.J., Arnold S., Brandt U. and Nijtmans L. 2016. Cell Metab, in press).

Reviewer #3 gave this extended explanation because the authors depended heavily on Moreno-Lastres proposal in interpreting their findings; it may be better to be cautious and explore other potential interpretations.

Results and Discussion:

Subsection “Isolation of mitochondrial supercomplexes solubilized with PCC-a-M”, second paragraph: The authors conclude that PCC-a-M does not allow detection of SC: I_1_III_2_ and infer that this SC is not observed because it is preserved as I_1_III_2_IV_1_. The authors should be aware that different detergents generate different micelles containing the SCs and modify their electrophoretic motility. In this sense Figure 1—figure supplement 1 does not help very much, since CIV is not detected in the first dimension in some of the samples (Figure 1 and Figure 1—figure supplement 1) while it is detected in others (Figure 1—figure supplement 1), and in the second dimension almost all the CIV seemed to be present as free complex. In addition, the authors find I_1_III_2_ by cryo-electron microscopy analysis in sufficient quantity to propose its structure (subsection “Defined protein-protein contacts”, first paragraph). They justify that observation by assuming that I_1_III_2_ is an artefact generated in vitro by cleavage of I_1_III_2_IV_2_ (in the second paragraph of the aforementioned subsection). This could be the explanation, but it is also possible that they co-purified both, but their detergent did not resolve them in a gel. In fact, they do not report the observation of free CIV that has to be present if the cleavage happened after purification. Finally, if the authors are interested in identifying properly the different components of the bands obtained with the new detergent they should use immunodetection approaches rather than Coomassie staining. There are good antibodies for that purpose.

Figure 3 shows three proteins: CI-B14.7, CIII-Subunit 8, and a third in yellow (CIV) that remains unidentified. Please identify it. Does it participate in the interaction between complexes I and IV or III and IV?

Subsection “Defined protein-protein contacts”, first paragraph. The minimum number of expected TMHs for the respirasome I_1_III_1_IV_1_ is 131 but the authors report only 114. Which ones are they unable to detect and why?

The authors report: "Several of the classes obtained (≈ 30% of the initial dataset) have no density for the NADH:dehydrogenase module of CI or even lack the whole matrix arm". Contrary to the argument given for the unexpected observation of I_1_III_2_ without IV, they do not consider these new classes as an artefact of the manipulation. Rather they propose that represent intermediates of supercomplex assembly, potentially in agreement with those postulated by Moreno-Lastres et al. (subsection “Conformational and compositional variability”, last paragraph). The model of for the assembly of CI and SCs proposed by Moreno-Lastres has been questioned, however (see above). Moreover, two different groups proposed that CI damage occurred under natural conditions at the NADH:DH module, and that this can be replaced without the elimination of the rest of the complex (Lazarou, et al. 2007. Mol Cell Biol 27, 4228-4237. Dieteren, et al. 2012 J. Biol. Chem 287, 41851-41860). If so, the observed structures could also be consistent with the prediction of Lazarou and Dieteren and might have come from degradation rather than as assembly intermediates. The data in this manuscript cannot determine whether those structures are natural or artefactual and whether they might represent intermediates of synthesis or degradation. The reviewer's suggestion is to present either all the potential alternative explanations or to simply remove the discussion of the nature of this structures. In particular, the suggestion that complex III acts as an assembly factor for CI (subsection “Supercomplex assembly, complex I stability and human health”, first paragraph) is not justified by the results presented nor by most of the recent literature.

The authors state that their proposed model of functional asymmetry and electron flow in CIII: "does not imply a partitioning of the quinol pool or substrate channeling". However, they also state: "A quinol diffusing through the lipid bilayer from complex I to complex III would thus encounter the active monomer first. The linear arrangement of complex I, the active complex III monomer and complex IV in the respirasome seems to be particularly favourable for efficient substrate and electron transfer." This is confusing, because the statements appear to be contradictory. On the one hand, the authors propose that the respirasome has a particular defined structure that favors oxidization of quinol not only the by CIII incorporated into the respirasome but also by a defined monomer. In the other, they state that this does not imply a kind of channeling. The reviewer believes that their proposed model entails preferential reduction of the ubiquinol within the respirasome rather than a more stochastic picture.

Subsection “Supercomplex assembly, complex I stability and human health”, second paragraph: The statement that deficiency in CIV does not have strong adverse effect on CI stability is incorrect (Diaz et al. 2006- Mol Cell Biol 26, 4872-4881& Vempati, et al. 2009. J. Biol. Chem. 284, 4383-4391).

Comments from reviewer #3 regarding comparison with the other published structures:

One of the two published structures found also two alternative conformations for the respirasome – one similar to the authors' Class 1 and the other more similar to Class 2. Two features characterize the differences between the two conformations: the twist in the position of CIII and the absence of contacts between CIII and CIV. In the Class 2 structure presented here, there is a similar twist in the relative position of CIII, but the authors do not state whether if the interaction between CIII and CIV is also absent. A comment on the similarities and differences between the two classes of structures of the respirasome reported here and published by Sazanov group would be of interest.

The Yang and Sazanov structures agree in the protein-protein contacts. The contacts proposed here are substantially similar but there are discrepancies. In particular, according to the structure proposed here, the interface between CIII and CIV is formed by CIII subunits 1 (UQCR1) and 9 (UQCR10) with COX7A1 of CIV, while Yang and Sazanov both say that CIII UQCR1 and UQCR 11 contact CIV subunit COX7A1. Is there an explanation for the discrepancy?

The Sazanov structure would permit an asymmetric flux of electrons within CIII in the respirasome. The present manuscript also presents an interesting model. It would be of great interest to discuss both, their similarities and their differences.

---

## [Author Response]

*[…] Summary and specific request:*

*This manuscript reports the structure of the bovine respirasome at 9 Å resolution. Its principal conclusion concerns a functional asymmetry: one of the iron-sulfur clusters is docked in such a way that electrons cannot be transferred, because the distance to the partner is too great. The authors conclude that one of the two is inactive. The structure is well determined at the stated resolution, but two structures at much higher resolution appeared after this manuscript was submitted. The reviewers request a revised manuscript that places the current conclusions in the context of those reports. In particular, the authors should address the likelihood of a different interpretation – some sort of alternating access model in which only one subunit can access the transfer site at a time, while the other docks in the observed, autoinhibited position. That is, the statement that the resolved Rieske center is immobile and therefore not functional is one possibility, but just because it has a fixed orientation in this structure of the inactive respirasome does not mean it cannot move in the active respirasome.*

We have added a comparison with the ovine and porcine structures at the end of the Discussion. The main map classes are very similar and the assignment of interacting subunits is essentially the same in all three structures. There are differences between the minor map classes of the ovine and our bovine supercomplexes, which may be either species-specific or purification dependent. It is likely that these minor particle classes are due to destabilization of the respirasomes.

The possibility of an alternating activation of complex III monomers is now discussed. A detailed explanation of why such a model disagrees with our structure is provided. The recent structures at higher resolution have both iron-sulfur domains resolved and do not provide further insights into the electron and substrate flow in the respirasome. The model proposed by Letts et al., (2016) is discussed in the context or our map.

*The following detailed comments may help with the revision.*

*Reviewer #2:*

*It is strange to show an FSC with phases randomized beyond 20 Å. Use of a phase randomized FSC to correct for effects from masking of maps is excellent, but the corrected FSC should be shown, not the phase randomized FSC.*

The FSC with randomized phases was removed from the graph. Only the corrected FSC is shown now.

*Reviewer #3:*

*Introduction:*

*First paragraph: The entrance of electrons in the mitochondrial electron transfer chain is branched, therefore CI is the largest complex, but cannot be called the first, because other enzymes that deliver electrons to CoQ are equally first in their branch. Please correct.*

The statement has been removed.

*First paragraph: It should be made clear that mammalian CI comprises 44 different subunits, one being duplicated. Quoting Carrol et al. 2002 could be misleading, since in this reference it was considered that CI comprises 45 different subunits. Latter it was demonstrated and generally accepted that NDUFA4 is not a CI subunit. It is advisable to clarify and to use the correct references.*

The number of CI subunits is now clearly stated and we cite Vinothkumar et al. 2014, where the SDAP subunit was shown to be duplicated:

“Mammalian complex I comprises 44 different subunits, including two copies of subunit SDAP, and therefore consists of a total of 45 subunits (Vinothkumar et al., 2014).”

*First paragraph: The number of mammalian CIV subunits has been proposed to be 14.*

The number of CIV subunits has been corrected and Balsa et al. 2012 is cited for identification of the 14^th^ subunit (NDUFA4):

“Mammalian complex IV has three core subunits (COX1, COX2 and COX3) and 14 subunits in total (Kadenbach et al., 1983; Balsa et al., 2012).”

*Third paragraph: The issue of CoQ partitioning is still hotly debated. The authors correctly mention that Blaza et al. 2014 disagree with this idea but forget to mention that those authors' experimental setup was rigorously contested in two recent reports (Enriquez 2016. Annu. Rev. Physiol. 78, 533-561 & Lenaz et al., 2016. Biochim Biophys Acta 1857, 991-1000). This is relevant because the very interesting proposal by Sousa et al. for the mechanism of electron transfer within the respirasome agrees with the interpretation of Enriquez and Lenaz rather than that of Blaza.*

We now provide a more comprehensive description of the current knowledge on substrate channeling by referring to the views of Enríquez (2016) and Lenaz et al. (2016) in the revised Discussion:

“Substrate channeling within the respirasome has been extensively debated. Results inconsistent with the existence of distinct quinone pools and substrate channeling have been reported (Blaza et al., 2014), but substrate channeling between complexes I and III is supported by most studies of metabolic flux control (Bianchi et al., 2003; Lapuente-Brun et al., 2013) and in recent reviews discussing the results of Blaza et al. (Enríquez, 2016; Lenaz et al., 2016).”

*Third paragraph: The best experimental support for the role of supercomplexes in minimizing ROS production was probably in the paper by Maranzana (Maranzana et al. 2013. Antioxid Redox Signal 19, 1469-1480), which is not quoted.*

We thank the reviewer for this reference, which is now cited in the fourth paragraph of the Introduction.

*Third paragraph: The role of supercomplexes in stabilizing CI is noted in the Introduction, but in a way that may not reflect the current view of this question. The historical origin of this proposal is as follows: in 2004 two groups independently proposed that CI stability was compromised in the absence of CIII (Acin-Perez et al., 2004 & Schägger et al. 2004). Latter, the group of Carlos Moraes demonstrated that ablation of CIV or cyt c, also compromised the stability of CI (Diaz et al. 2006- Mol Cell Biol 26, 4872-4881 & Vempati, et al. 2009. J. Biol. Chem. 284, 4383-4391). More recently it was demonstrated that CI can be assembled functionally in the absence of CIII, CIV or cyt c by recovering the re-oxidation of CoQ (Guarás, 2016. Cell Reports 15, 197-209). This series of reports led to the conclusion that assembly of the individual respiratory complexes can proceed independently and requires neither the presence of other complexes nor interaction with them. In line with this conclusion, kinetic analysis showed that the formation of complexes happened before the formation of supercomplexes (Acín-Peréz et al. 2008. Mol. Cell 32, 529-539). This result does not preclude, however, that incomplete CI or CIII could interact with other complexes and form supercomplexes, as reported by several groups (Marques, I., 2007. Eukaryotic Cell 6, 2391-2405). In 2012 it was proposed that the assembly of the NADH-DH module of CI require that formation of a non-functional respirasome between partially assembled CI with CIII and CIV (Moreno-Lastres et al. 2012. Cell Metab.15, 324-335). This proposal contradicted that of (Acín-Peréz et al. 2008. Mol. Cell 32, 529-539). A very recent reevaluation concluded that CI, CIII and CIV are fully assembled before they associate into supercomplexes, refuting the proposal by Moreno-Lastres (Guerrero-Castillo S, Baertling F, Kownatzki D., Wessels H.J., Arnold S., Brandt U. and Nijtmans L. 2016. Cell Metab, in press).*

*Reviewer #3 gave this extended explanation because the authors depended heavily on Moreno-Lastres proposal in interpreting their findings; it may be better to be cautious and explore other potential interpretations.*

We thank the reviewer for bringing this point to our attention. Our revised Introduction now provides a more detailed and balanced description of the stability of complex I and how it depends on other respiratory complexes:

“Several mitochondrial disorders are associated with impaired respirasome formation. […] When ROS production by RET was inhibited, complex I levels were restored (Guaras et al., 2016).”

We also mention the recent results on the assembly pathway of supercomplexes:

“However, recent complexome profiling studies failed to detect supercomplexes containing immature complex I, suggesting that respirasomes form by association of fully assembled component complexes (Guerrero-Castillo, 2016).”

*Results and Discussion:*

*Subsection “Isolation of mitochondrial supercomplexes solubilized with PCC-a-M”, second paragraph: The authors conclude that PCC-a-M does not allow detection of SC: I_1_III_2_ and infer that this SC is not observed because it is preserved as I_1_III_2_IV_1_. The authors should be aware that different detergents generate different micelles containing the SCs and modify their electrophoretic motility. In this sense Figure 1—figure supplement 1 does not help very much, since CIV is not detected in the first dimension in some of the samples (Figure 1 and Figure 1—figure supplement 1) while it is detected in others (Figure 1—figure supplement 1), and in the second dimension almost all the CIV seemed to be present as free complex. In addition, the authors find I_1_III_2_ by cryo-electron microscopy analysis in sufficient quantity to propose its structure (subsection “Defined protein-protein contacts”, first paragraph). They justify that observation by assuming that I_1_III_2_ is an artefact generated in vitro by cleavage of I_1_III_2_IV_2_ (in the second paragraph of the aforementioned subsection). This could be the explanation, but it is also possible that they co-purified both, but their detergent did not resolve them in a gel. In fact, they do not report the observation of free CIV that has to be present if the cleavage happened after purification. Finally, if the authors are interested in identifying properly the different components of the bands obtained with the new detergent they should use immunodetection approaches rather than Coomassie staining. There are good antibodies for that purpose.*

The reviewer’s point regarding the possible effects of detergents on resolving membrane protein complexes on a gel is well taken. However, our negative stain EM analysis clearly shows that virtually all particles of the sample purified in PCC-a-M contain complex IV, whereas a substantial portion lacks this complex in the digitonin-solubilized sample (Figure 1—figure supplement 1). We are therefore confident that our conclusions regarding the presence or absence of complex IV in the gels are correct.

CIV is quite small (~200 kDa) compared to the respirasome (1.7 MDa) and therefore hard to see on cryo-EM grids, especially at a defocus optimized for larger particles, which would make small particles even more difficult to detect. We did not pick particles of this small size for our analysis, which explains the absence of class averages of free CIV.

*Figure 3 shows three proteins: CI-B14.7, CIII-Subunit 8, and a third in yellow (CIV) that remains unidentified. Please identify it. Does it participate in the interaction between complexes I and IV or III and IV?*

The point of Figure 3 is to demonstrate the interaction between CI and CIII. It shows a segment of the respirasome map with complexes colored in blue (CI), green (CIII) and yellow (CIV), as in Figure 2 (and as indicated in the figure legend). Darker shades of blue and green are used to indicate the subunits involved in CI/CIII interaction in the membrane and on the membrane surface.

Other points of interaction between the complexes are mentioned explicitly in the Results under “Defined protein-protein contacts”. The map section in Figure 3 does show one of these contacts (between CI-ND5 and CIV-COX7C) but we prefer to keep the focus of this figure on the interactions between complex I and III as the most extensive contact region in the respirasome.

*Subsection “Defined protein-protein contacts”, first paragraph. The minimum number of expected TMHs for the respirasome I_1_III_1_IV_1_ is 131 but the authors report only 114. Which ones are they unable to detect and why?*

Thank you for pointing this out. We simply forgot to count the TMHs of the supernumerary CI subunits. In the revised manuscript this error has been corrected. The total number of TMHs is actually 132 (78 in CI, 26 in the CIII dimer and 28 in CIV). Our map resolves most TMHs in the respirasome, but not all, due to variations in local resolution (see Figure 2—figure supplement 1). In particular, all TMHS from CI are resolved, whereas some in complex III and most in complex IV are not. This concurs with our observation that complex IV is less firmly held in the respirasome than complex III.

*The authors report: "Several of the classes obtained (≈ 30% of the initial dataset) have no density for the NADH:dehydrogenase module of CI or even lack the whole matrix arm". Contrary to the argument given for the unexpected observation of I_1_III_2_ without IV, they do not consider these new classes as an artefact of the manipulation. Rather they propose that represent intermediates of supercomplex assembly, potentially in agreement with those postulated by Moreno-Lastres et al. (subsection “Conformational and compositional variability”, last paragraph). The model of for the assembly of CI and SCs proposed by Moreno-Lastres has been questioned, however (see above). Moreover, two different groups proposed that CI damage occurred under natural conditions at the NADH:DH module, and that this can be replaced without the elimination of the rest of the complex (Lazarou, et al. 2007. Mol Cell Biol 27, 4228-4237. Dieteren, et al. 2012 J. Biol. Chem 287, 41851-41860). If so, the observed structures could also be consistent with the predicition of Lazarou and Dieteren and might have come from degradation rather than as assembly intermediates. The data in this manuscript cannot determine whether those structures are natural or artefactual and whether they might represent intermediates of synthesis or degradation. The reviewer's suggestion is to present either all the potential alternative explanations or to simply remove the discussion of the nature of this structures. In particular, the suggestion that complex III acts as an assembly factor for CI (subsection “Supercomplex assembly, complex I stability and human health”, first paragraph) is not justified by the results presented nor by most of the recent literature.*

The reviewer is correct in that our respirasome maps cannot differentiate between complex I assembly and disassembly intermediates. Strong evidence for two distinct assembly models of the respirasome has been reported and we refer to these models in the revised Introduction. The revised manuscript still discusses our observations in the context of the model proposed by Moreno-Lastres 2012, but we now point out that the alternative model of Guerrero-Castillo (2016) does not support assembly intermediates as those described (subsection “Conformational and compositional variability”, second paragraph). In addition, we now discuss the model for direct exchange of CI subunits that suffer most from oxidative damage, which would obviate the need for de novo synthesis of the whole assembly (in the last paragraph of the aforementioned subsection), as proposed by Lazarou et al. (2007) and Dieteren et al. (2012). We discuss in particular how this model might relate to the 3D class averages we observe.

We explicitly do not discount the possibility that the incomplete classes result from partial disassembly of the respirasome during protein purification or cryo-EM grid preparation. In fact, we already pointed this out in the original manuscript. In the revised manuscript we emphasize this before mentioning any possible assembly, disassembly or recycling intermediates:

“Several of the classes obtained (~30% of the initial dataset) have no density for the NADH:dehydrogenase module of complex I, or even lack the whole matrix arm (Figure 1—figure supplement 2). While these classes might reflect a partial loss of this module during purification or cryo-EM grid preparation, we cannot rule out the possibility that they represent biologically relevant assembly, disassembly or recycling intermediates of the respirasome.”

*The authors state that their proposed model of functional asymmetry and electron flow in CIII: "does not imply a partitioning of the quinol pool or substrate channeling". However, they also state: "A quinol diffusing through the lipid bilayer from complex I to complex III would thus encounter the active monomer first. The linear arrangement of complex I, the active complex III monomer and complex IV in the respirasome seems to be particularly favourable for efficient substrate and electron transfer." This is confusing, because the statements appear to be contradictory. On the one hand, the authors propose that the respirasome has a particular defined structure that favors oxidization of quinol not only the by CIII incorporated into the respirasome but also by a defined monomer. In the other, they state that this does not imply a kind of channeling. The reviewer believes that their proposed model entails preferential reduction of the ubiquinol within the respirasome rather than a more stochastic picture.*

The statement has been removed and the fact that our structure agrees with quinone channeling is now stated clearly in the revised manuscript:

“The linear arrangement of complex I, the active complex III monomer and complex IV in the respirasome seems to be particularly favourable for efficient substrate and electron transfer. […] The asymmetry observed for complex III in our supercomplex structure is in agreement with these results and provides further support for electron transfer by substrate channeling in the respirasome.”

*Subsection “Supercomplex assembly, complex I stability and human health”, second paragraph: The statement that deficiency in CIV does not have strong adverse effect on CI stability is incorrect (Diaz et al. 2006- Mol Cell Biol 26, 4872-4881& Vempati, et al. 2009. J. Biol. Chem. 284, 4383-4391).*

The reviewer is correct; CIV deficiencies can induce a secondary loss of CI. The paper cited by us (D’Aurelio 2006) mentions, however, that high mutation levels of CIV are required to induce a secondary loss of CI in human cells. Such high mutation levels are not of practical relevance in our case and we have therefore removed this statement from the revised manuscript.

*Comments from reviewer #3 regarding comparison with the other published structures:*

*One of the two published structures found also two alternative conformations for the respirasome – one similar to the authors' Class 1 and the other more similar to Class 2. Two features characterize the differences between the two conformations: the twist in the position of CIII and the absence of contacts between CIII and CIV. In the Class 2 structure presented here, there is a similar twist in the relative position of CIII, but the authors do not state whether if the interaction between CIII and CIV is also absent. A comment on the similarities and differences between the two classes of structures of the respirasome reported here and published by Sazanov group would be of interest.*

*The Yang and Sazanov structures agree in the protein-protein contacts. The contacts proposed here are substantially similar but there are discrepancies. In particular, according to the structure proposed here, the interface between CIII and CIV is formed by CIII subunits 1 (UQCR1) and 9 (UQCR10) with COX7A1 of CIV, while Yang and Sazanov both say that CIII UQCR1 and UQCR 11 contact CIV subunit COX7A1. Is there an explanation for the discrepancy?*

*The Sazanov structure would permit an asymmetric flux of electrons within CIII in the respirasome. The present manuscript also presents an interesting model. It would be of great interest to discuss both, their similarities and their differences.*

In the revised manuscript, we have added a comparison of our maps and the recently published structures of porcine and ovine respirasome to the end of the Discussion (subsection “Supercomplex assembly, complex I stability and human health”, third paragraph). A detailed comparison with the porcine map was unfortunately not possible since neither the map nor the model coordinates are available at this point. We would like to call your attention to the following points:

Inspection of the released coordinates for the tight conformation of the ovine respirasome reveal distances between subunit 9 and COX7A1 of 16 Å, which is longer than in our bovine complex and would be incompatible with the interaction that we propose. This could be due to differences of the species, preparation or resolution.

Re-examination of our map in the light of the other published structures revealed that subunit 10 of CIII comes to within 5 Å of the N terminus of COX7A1, and thus also contributes to CIII/CIV interactions. We added this to the list of subunits participating in protein-protein contacts in the respirasome (subsection “Defined protein-protein contacts”, last paragraph).

A comparative analysis of our model with the one from Letts et al. (2016) is provided in the revised Discussion (last paragraph). The models agree on the participation of the central monomer from complex III in quinone oxidation, in our case based on the asymmetry of the iron-sulfur domains, in their case based on the proximity of the outer iron-sulfur domain to complex IV that might restrict its movement. Letts et al. further propose that quinone reduction should take place in the outer monomer. We do not find evidence in any of the structures that can support or contradict this hypothesis.